# Effects of Cultivation Management on Pearl Millet Yield and Growth Differed with Rainfall Conditions in a Seasonal Wetland of Sub-Saharan Africa

Yoshihiro Hirooka [1], Simon K. Awala [2], Kudakwashe Hove [2], Pamwenafye I. Nanhapo [2] and Morio Iijima [1,*]

[1] Graduate School of Agriculture, Kindai University, Nara 631-8505, Japan; hirooka@nara.kindai.ac.jp
[2] Faculty of Agriculture and Natural Resources, University of Namibia, Windhoek 13301, Namibia; sawala@unam.na (S.K.A.); khove@unam.na (K.H.); pnanhapo@unam.na (P.I.N.)
* Correspondence: iijimamorio@nara.kindai.ac.jp; Tel.: +81-742-43-7209

**Abstract:** The production of pearl millet (*Pennisetum glaucum (L.) R.Br.*) is important in Namibia, in sub-Saharan Africa, owing to the prevailing low precipitation conditions. Most fields supporting crop production in northern Namibia are located in a network of seasonal wetlands. The aim of the present study was to evaluate the effects of ridging and fertilizer application on the yield and the growth of pearl millet in the seasonal wetlands under different rainfall conditions. The study was conducted for two years (2017–2018) in the experimental fields in northern Namibia, and yield, yield components, and growth parameters were evaluated in relation to the application of different fertilizers (manure and mineral) with and without ridge-furrows. Manure fertilizer application presented the highest yield in 2018, whereas mineral fertilizer application showed the highest yield in 2017. The proportion of rainfall was the highest during the mid-growth period in 2017, and the reproductive stage in 2018. Thus, pearl millet plants under manure fertilization overcame damage resulting from waterlogging stress during the seed setting stage by improving the soil and plant nutrient conditions. In contrast, the plants under mineral fertilization were more tolerant to large amounts of rain during the mid-growth period. In this study, yield was mainly determined by total dry weight, and it was closely related to panicle density in both years. Therefore, we concluded that fertilizer application, including additional fertilizer based on the growth diagnostic, could be important for improving crop production in seasonal wetlands.

**Keywords:** climate change; drought-tolerant crop; irregular flooding; pearl millet; seasonal wetland; soil fertility

## 1. Introduction

Challenges associated with sustainable food production are being intensified by climate change, which is expected to have relatively higher effects in semi-arid and arid regions than in any other region. Human activities in these areas lead to land degradation, biodiversity reduction, and increased water scarcity [1,2]. Semi-arid sub-Saharan Africa is considered to be particularly vulnerable to the effects of climate change because of its high dependence on agriculture and natural resources under the prevailing conditions of low precipitation [3]. Consequently, the production of drought-tolerant crops, such as pearl millet (*Pennisetum glaucum (L.) R.Br.*), is important in this region [4,5].

Namibia is a semi-arid sub-Saharan African country, where the amount and distribution of rainfall vary considerably across different years [6]. Extreme droughts and floods have become frequent, particularly affecting the semi-arid northern regions of the country [7,8]. Most of the pearl millet production in northern Namibia occurs in a network of seasonal wetlands originating in the upper catchments of the neighboring countries, Angola and Zambia. Pearl millet is a staple food for more than 60% of the population in Namibia, and water accumulating in these seasonal wetlands, which are the lower

sections of field slopes, influences the yield gradient [9]. These seasonal wetlands have the potential to produce high yields of pearl millet in drought years because of the suitable water environment and soil fertility [10]. Moreover, cultivation in the seasonal wetlands could improve the agricultural production system in northern Namibia [11].

Iijima et al. [12] reported that floodwater in the Angolan plateau creates a vast seasonal wetland covering approximately 800,000 ha during the rainy season in the semi-arid areas of southern Angola and northern Namibia. Ridge-furrow tillage represents an efficient approach in these contrasting and unpredictable environments. This cultivation method modifies the soil-moisture environment by increasing aeration in the ridges and retaining moisture in the furrows. Therefore, planting drought-tolerant crops on ridges and furrows might help to buffer the loss in crop yield attributable to droughts or floods. In addition, ridging is becoming popular in areas where the frequency of flooding has increased [13]. Typical tillage methods in such areas include moldboard plowing with donkeys or oxen. Recently, four-wheel tractors mounted with various types of harrows have become a common alternative.

Previous studies have indicated that pearl millet yield increases with mineral fertilizer application, but it is not widely adopted because of the limited availability and high cost of fertilizers [14]. Alternatively, cattle manure can be applied in these areas to supply plants with nutrients and to alleviate waterlogging stress by improving the soil's physical properties and drainage [15]. Cattle manure generally decreases agronomic and economic risks by supporting crop production in sandy soils [16]. Ridging is also an efficient management practice for millets in semi-arid area [17–21]. In addition, the combination of ridging with other cultivation management strategies, including cultivar selection, improved millet production [18,22,23]. However, the interactions between rainfall conditions and cultivation management strategies, such as ridging and fertility management based on manure and mineral fertilizers, and their impact on the yield and growth of pearl millet have not been quantified in seasonal wetlands, which can serve as efficient arable land for pearl millet cultivation under highly variable rainfall conditions.

The objective of this study was to evaluate how ridging and the application of different fertilizers influence the yield and growth of pearl millet in a seasonal wetland in sub-Saharan Africa and to quantify their effects under different rainfall conditions. We hypothesized that the optimum cultivation management would differ with rainfall conditions in these areas.

## 2. Materials and Methods

### 2.1. Study Site

The present study was conducted in 2017 and 2018 in the experimental fields of the University of Namibia, Ogongo Campus (17°41′ S, 15°18′ E, 1109 m above sea level), Namibia. The recent time profile of rainfall in the experimental region (Omahenene; 17°27′ S, 14°47′ E) is shown in Figure 1. The experiments were conducted on the lower part of a slope in the field to simulate cultivation in small, seasonal wetlands. An experimental field was set up, which was 80 m × 160 m in size, with an artificial slope of 1/80 [24,25]. Weather data were obtained from the Southern African Science Service Centre for Climate Change and Adaptive Land Management (SASSCAL) website (http://www.sasscalweathernet.org/weatherstat_monthly_we.php, accessed date: 10 March 2021). The soil in the experimental site was classified as sandy [26]; the soil in the area (arid zone) shows lower infiltration and percolation rates than tropical and temperate zones, and at the surface, it tends to hold excess moisture [27].

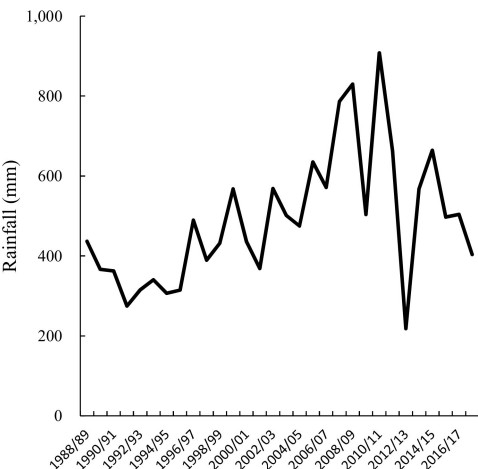

**Figure 1.** Annual rainfall trend in northern Namibia (Omahenene, 17°27′ S, 14°47′ E) over a 30-year period.

### 2.2. Plant Materials and Experimental Design

On 24 January 2017 and 5 January 2018, a few days after mineral fertilizer application, seeds of pearl millet variety 'Okashana 2' were sown at 3 cm depth using an 80 cm inter-row spacing and a 40 cm intra-row spacing, with a planting density of 3 plants per m$^2$ (standard planting density in the area). The seeds were obtained from the Namibian Seed Company.

The experiments were arranged in a split-plot design with conventional/flat-bed (Cf) and ridge-furrow (R/F) tillage assigned to the main plots, and the three fertilization treatments: no fertilization, cattle manure fertilization (dry basis; 10 t ha$^{-1}$), and NPK mineral fertilization (30:45:30 kg ha$^{-1}$ N/P$_2$O$_5$/K$_2$O) assigned to the subplots. The amount was determined based on the local recommendation in the target area. The same plot treatment allocations were used in the second year as in the first. Both manure and mineral fertilizer were surface-applied twice before starting the 2017 and 2018 experiments based on common practice by farmers in the targeted area. Manure fertilizer was applied four weeks before sowing. Almost half of the manure was organic matter, and the total levels of N, P, and K were 9.5, 1.6, and 4.7 g kg$^{-1}$, respectively [15]. The experiments in both years were carried out with three replications.

The Cf treatment was implemented with a rotary plow to prevent the fertilizers applied during the previous season from drifting into the adjacent plots. The R/F treatment was implemented as inter-ridge furrows using a chisel plow to loosen the soil at the bottom of the furrows and facilitate the subsequent planting operations. Ridge-furrow tillage plots were constructed using a ridger. The ridges were 30–40 cm high, with an inter-ridge spacing of 1.6 m. After constructing the ridge-furrow plots, the main-plots were divided into individual fertilizer sub-plots of 156.8 m$^2$ (14.0 m × 11.2 m). In the R-F plots, pearl millet seeds were sown on ridges and in furrows. A split-plot design was used to assess the synergistic effects of soil tillage and fertilization on the growth and yield of pearl millet.

Pesticides were not used to control insects or diseases, as the incidence and damage caused by such pests were negligible. Weeding was carried out manually, once a week after thinning. Bird-scarers were installed during the seed setting stage to minimize grain damage from the grain setting stage to harvest. The land was managed by a rotary plow in Cf treatment and furrows, by hand hoeing in the ridge between the first- and second-year crops.

### 2.3. Measurements

Crop yield was evaluated after the plants reached maturity, which was on 28 April in 2017 and 24 April in 2018. At harvest, the sampling area of each sub-plot was 49.9 m$^2$ (5.2 m × 9.6 m). The number of panicles per m$^2$, number of grains per panicle, 1000-grain weight, plant height, and panicle length were determined using 24 randomly selected plants (in

R/F treatment, R = 12 plants and F = 12 plants). Aboveground total dry weight (TDW) was estimated from the plants. All panicles within the sampling area were harvested and air-dried for more than 3 weeks before hand-threshing. The grains were weighed, and the moisture content measured using a grain-moisture tester (PM-830-2; Kett, Tokyo, Japan). Grain weight was adjusted to 14% moisture content to estimate grain yield. Harvest index (HI) was calculated as: (yield/TDW).

To evaluate the soil-water environment during the rainfall period, soil-moisture content in the topsoil layer (10 cm to 20 cm) and the layer below (30 cm to 40 cm) was measured with three replications per plot at 7-day intervals from 57 to 85 days after seeding in 2018 using ML2x Theta Probe (Delta-T Devices Ltd., Cambridge, UK). In addition, to evaluate the physiological activity during the seed setting stage, the chlorophyll content (SPAD value) of the uppermost fully expanded leaf of three plants per plot was measured at 71 and 92 days after seeding in 2018 using the SPAD-502 meter (Minolta Camera Co., Ltd., Osaka, Japan).

After completing the two-year experiment, soil samples were collected from each sub-plot. Three sub-samples were collected per plot (ridges and furrows were sampled separately). The sub-samples from each plot were thoroughly mixed to obtain composite samples from Cf, ridge, and furrow areas. All soil samples were air-dried, ground, and passed through a two-millimeter sieve before analysis. Nitrate nitrogen was determined using the cadmium-reduction method [28]. Available phosphorus and potassium were determined using the ascorbic acid-reducing method [29] and the tetraphenylboron method [30], respectively. Soil pH was measured in 1:2.5 suspensions of soil and water, and soil organic carbon (OC) was determined using the potassium dichromate-oxidation method [31]. Electrical conductivity (EC) was determined by from the supernatant of a 1:2.5 soil:water suspension with a conductivity meter.

The effects of tillage and fertilization treatments on yield, yield components, soil characteristics, soil-moisture content, and SPAD value were analyzed using two-way ANOVA. All statistical analyses were conducted using R version 3.4.0 (R Core Team (2017)).

## 3. Results

In 2017 and 2018, the total rainfall during the crop growth periods was 254.5 and 189.6 mm, respectively (Table 1). Even in years with low rainfall, e.g., in 2018, pearl millet reached maturity in the experimental field. Signs of surface water movement and short-term flooding of the crop were observed in early March during the reproductive growth stage in 2017 and in early April during the seed setting stage in 2018.

**Table 1.** Daily mean temperature ($^\circ$C), solar radiation (MJ m$^{-2}$ d$^{-1}$) and monthly precipitation (mm) during the crop growth periods.

|  | Temperature ($^\circ$C) | | | Solar Radiation | Precipitation |
|---|---|---|---|---|---|
|  | **Mean** | **Min** | **Max** | **(MJ m$^{-2}$ d$^{-1}$)** | **(mm)** |
| 2017 |  |  |  |  |  |
| January | 24.2 | 18.6 | 30.4 | 21.3 | 36.8 |
| February | 24.5 | 19.7 | 30.2 | 19.8 | 76.6 |
| March | 24.0 | 19.9 | 29.3 | 20.7 | 118.2 |
| April | 22.7 | 16.9 | 29.7 | 20.4 | 22.9 |
| Ave. | 23.8 | 18.9 | 29.8 | 20.4 | total 254.5 |
| 2018 |  |  |  |  |  |
| January | 27.0 | 20.3 | 34.0 | 26.1 | 0.8 |
| February | 26.0 | 19.4 | 33.1 | 25.2 | 16.4 |
| March | 23.9 | 19.1 | 30.2 | 20.5 | 63.6 |
| April | 23.1 | 18.2 | 29.2 | 18.3 | 108.8 |
| Ave. | 25.0 | 19.3 | 31.7 | 22.6 | total 189.6 |

Grain yield in 2017 ranged from an average of 0.49 t ha$^{-1}$ for the unfertilized treatment to 1.02 t ha$^{-1}$ where mineral fertilizer had been added (Table 2a). In 2018, the range in grain yields was from 2.0 t ha$^{-1}$ for the unfertilized treatment to 3.97 t ha$^{-1}$ for the manured treatment. In neither year was there a response in grain yield to tillage treatment (Table 2b). The HI averaged 0.26 in 2017 and 0.44 in 2018. The higher yield for fertilized treatments was through a higher panicle density, and more grains per panicle (in 2018). The 1000-grain weight was lower in the R/F treatment in 2017, and was not significantly affected by fertilizer treatment. Fertilizer was associated with an increase in plant height in 2017, while panicle length was unaffected by either treatment. Grain yield was most strongly correlated with maturity biomass, followed by panicle density (Table 3). The panicle density was significantly increased by fertilizer in both years, especially in mineral fertilizer. In turn, the grain density was not significantly increased by mineral fertilizer. The 1000-grain weight was significantly increased by manure fertilizer in both years.

**Table 2.** Averages and results of ANOVA for comparison of yield and yield components of pearl millet among tillage and fertilizer treatments.

| (a) 2017 | | | | | | | | |
|---|---|---|---|---|---|---|---|---|
| | **Grain Yield** (t ha$^{-1}$) | **Total Dry Weight** (t ha$^{-1}$) | **Harvest Index** | **Panicle Density** (m$^{-2}$) | **Grain Density** (10$^2$) | **1000-Grain Weight** | **Plant Height** (cm) | **Panicle Length** (cm) |
| Tillage | | | | | | | | |
| Cf | 0.74 | 2.90 | 0.26 | 5.8 | 0.73 | 17.7 | 179 | 19.9 |
| R/F | 0.71 | 2.73 | 0.26 | 6.7 | 0.60 | 15.8 | 177 | 19.4 |
| LSD ($p < 0.05$) | 0.26 | 0.87 | 0.06 | 1.7 | 0.11 | 1.3 | 11 | 1.1 |
| Fertilizer | | | | | | | | |
| None | 0.49 | 2.27 | 0.22 | 4.7 | 0.64 | 15.7 | 168 | 19.2 |
| Manure | 0.67 | 2.68 | 0.25 | 6.6 | 0.59 | 17.2 | 179 | 19.9 |
| Mineral | 1.02 | 3.49 | 0.29 | 7.5 | 0.77 | 17.3 | 187 | 19.7 |
| LSD ($p < 0.05$) | 0.16 | 0.88 | 0.06 | 1.6 | 0.13 | 1.9 | 9 | 1.4 |
| ANOVA | | | | *p* value | | | | |
| Tillage | 0.601 | 0.572 | 0.824 | 0.078 | 0.033 * | 0.005 ** | 0.471 | 0.356 |
| Fertilizer | 0.000 *** | 0.016 * | 0.057 | 0.002 ** | 0.038 * | 0.084 | 0.004 ** | 0.515 |
| Tillage*Fertilizer | 0.225 | 0.043 * | 0.105 | 0.085 | 0.430 | 0.920 | 0.387 | 0.280 |
| (b) 2018 | | | | | | | | |
| | **Grain Yield** (t ha$^{-1}$) | **Total Dry Weight** (t ha$^{-1}$) | **Harvest Index** | **Panicle Density** (m$^{-2}$) | **Grain Density** (10$^2$) | **1000-Grain Weight** | **Plant Height** (cm) | **Panicle Length** (cm) |
| Tillage | | | | | | | | |
| Cf | 2.79 | 7.01 | 0.40 | 12.0 | 1.36 | 17.1 | 170 | 19.9 |
| R/F | 3.13 | 6.64 | 0.47 | 12.1 | 1.63 | 16.6 | 169 | 19.4 |
| LSD ($p < 0.05$) | 0.30 | 1.84 | 0.07 | 1.8 | 0.37 | 1.0 | 8 | 1.1 |
| Fertilizer | | | | ; | | | | |
| None | 2.00 | 4.70 | 0.43 | 10.3 | 1.31 | 16.4 | 171 | 24.8 |
| Manure | 3.97 | 8.42 | 0.47 | 12.5 | 1.84 | 17.6 | 171 | 24.2 |
| Mineral | 2.90 | 7.36 | 0.39 | 13.3 | 1.33 | 16.5 | 167 | 25.7 |
| LSD ($p < 0.05$) | 0.39 | 1.05 | 0.10 | 1.5 | 0.38 | 1.0 | 10 | 1.1 |
| ANOVA | | | | *p* value | | | | |
| Tillage | 0.011 * | 0.408 | 0.093 | 0.892 | 0.078 | 0.292 | 0.784 | 0.207 |
| Fertilizer | 0.000 *** | 0.000 *** | 0.309 | 0.005 ** | 0.016 * | 0.049 * | 0.725 | 0.033 * |
| Tillage*Fertilizer | 0.112 | 0.804 | 0.655 | 0.306 | 0.560 | 0.577 | 0.642 | 0.342 |

Grain yield at 14% moisture content. Cf, conventional flat-bed tillage; R/F, ridge and furrow tillage; *, **, and *** indicate significant effects at 0.05, 0.01, and 0.001 probability levels, respectively.

**Table 3.** Pearson partial correlation coefficients for yield and growth characteristics in 2017 and 2018.

| | Yield | Total Dry Weight | Harvest Index | Panicle Density | Grain Density | 1000-Grain Weight | Plant Height | Panicle Length |
|---|---|---|---|---|---|---|---|---|
| Yield | | 0.78 ** | 0.39 | 0.74 ** | 0.50 * | 0.33 | 0.63 ** | 0.12 |
| Total dry weight | 0.78 ** | | −0.21 | 0.77 ** | 0.22 | 0.15 | 0.45 | −0.25 |
| Harvest index | 0.38 | −0.26 | | 0.05 | 0.49 * | 0.38 | 0.45 | 0.52 * |
| Panicle density | 0.71 ** | 0.67 ** | −0.32 | | −0.16 | 0.13 | 0.44 | −0.04 |
| Grain density | 0.43 | 0.28 | 0.67 ** | −0.25 | | 0.15 | 0.28 | 0.09 |
| 1000-grain weight | 0.42 | 0.56 * | −0.19 | 0.48 * | −0.12 | | 0.58 * | 0.49 * |
| Plant height | 0.01 | 0.08 | −0.09 | 0.31 | −0.36 | 0.37 | | 0.30 |
| Panicle length | 0.31 | 0.40 | −0.05 | 0.14 | 0.02 | 0.58 * | 0.66 ** | |

Above diagonal for the data in 2017 (*n* = 18); below diagonal for the data in 2018 (*n* = 18). * and ** indicate a significant correlation at 0.05 and 0.01 probability levels, respectively.

Soil N and K were significantly affected by tillage, fertilization and their interaction effects, and soil P was significant affected by fertilization and their interaction effects (Table 4). Soil N had the highest value in the ridge plot, and soil K had the highest value in the furrow plot. Mineral fertilization plots presented the highest soil P content, whereas the manure-fertilized plots showed the highest soil K content. The mineral fertilization plots showed the lowest soil N content. Manure fertilization significantly increased the soil pH. The soil OC, EC, and physical characteristics did not show significant tillage and fertilization effects, although the soil OC tended to be higher for the manure treatment.

**Table 4.** Residual effects of tillage and fertilizer treatments for soil characteristics (0–10 cm) in April 2018 after completion of the second year of the experiment.

| | N | P | K | OC | EC | Sand | Clay | Silt | pH |
|---|---|---|---|---|---|---|---|---|---|
| | (mg kg$^{-1}$) | (mg kg$^{-1}$) | (mg kg$^{-1}$) | (g kg$^{-1}$) | (ds m$^{-1}$) | (%) | (%) | (%) | |
| Tillage | | | | | | | | | |
| Cf | 8.3 | 25.1 | 41.1 | 2.70 | 0.70 | 64.5 | 25.1 | 8.9 | 6.9 |
| F | 7.9 | 24.5 | 49.6 | 2.56 | 0.70 | 67.0 | 23.0 | 10.3 | 7.1 |
| R | 10 | 21.9 | 33.8 | 2.81 | 0.71 | 67.4 | 25.5 | 5.9 | 6.6 |
| LSD (*p* < 0.05) | 1.4 | 18.1 | 12.4 | 0.35 | 0.04 | 6.9 | 7.2 | 4.5 | 0.8 |
| Fertilizer | | | | | | | | | |
| None | 9.4 | 14.2 | 48.2 | 2.44 | 0.68 | 64.4 | 26.2 | 7.8 | 6.7 |
| Manure | 9.1 | 13.5 | 44.1 | 2.96 | 0.71 | 64.4 | 25.4 | 8.9 | 7.4 |
| Mineral | 7.7 | 43.8 | 32.1 | 2.67 | 0.71 | 70.0 | 21.9 | 8.5 | 6.5 |
| LSD (*p* < 0.05) | 1.5 | 10.4 | 12.2 | 0.59 | 0.04 | 6.5 | 7.0 | 4.9 | 0.7 |
| ANOVA | | | | | *p* value | | | | |
| Tillage | 0.000 *** | 0.592 | 0.012 * | 0.245 | 0.858 | 0.639 | 0.767 | 0.223 | 0.274 |
| Fertilizer | 0.002 ** | 0.000 *** | 0.011 * | 0.082 | 0.222 | 0.183 | 0.477 | 0.901 | 0.023 * |
| Tillage*Fertilizer | 0.000 *** | 0.001 *** | 0.001 ** | 0.312 | 0.749 | 0.641 | 0.823 | 0.938 | 0.603 |

Cf, conventional flat-bed tillage; F, furrow plot; R, ridge plot. N, nitrate nitrogen; P, available P; K, exchangeable K; OC, organic carbon; EC, electrical conductivity; *, **, and *** indicate significance at 0.05, 0.01, and 0.001 probability levels, respectively.

The soil moisture content at different soil depths varied across tillage plots (Table 5). The furrow plots presented significantly higher soil moisture content at both depths, and the pattern was evident in the top soil layer. The soil moisture content in the ridge plots was similar to that in the flat plots in the top soil layer, and it was lower in the layer below. The manure-fertilized plots tended to have a lower soil moisture content than other plots, although this was not significant.

**Table 5.** Gravimetric soil moisture (g water/g dry soil) at depths of (a) 10–20 cm and (b) 30–40 cm during March 2018.

| (a) | | | | | |
|---|---|---|---|---|---|
| | **3-Mar** | **10-Mar** | **17-Mar** | **24-Mar** | **31-Mar** |
| Tillage | | | | | |
| Cf | 14.9 | 11.4 | 20.3 | 14.8 | 18.3 |
| F | 19.2 | 17.4 | 25.9 | 21.1 | 24.7 |
| R | 13.3 | 9.9 | 18.5 | 14.0 | 17.9 |
| LSD ($p < 0.05$) | 3.1 | 3.4 | 3.7 | 3.7 | 2.7 |
| Fertilizer | | | | | |
| None | 15.4 | 12.8 | 21.6 | 17.3 | 20.6 |
| Manure | 15.1 | 12.2 | 20.3 | 15.7 | 19.4 |
| Mineral | 16.9 | 13.6 | 22.8 | 17.0 | 21.0 |
| LSD ($p < 0.05$) | 4.0 | 4.7 | 4.8 | 4.9 | 4.1 |
| ANOVA | | | *p* value | | |
| Tillage | 0.001 *** | 0.001 ** | 0.000 *** | 0.001 *** | 0.000 *** |
| Fertilizer | 0.383 | 0.693 | 0.211 | 0.61 | 0.496 |
| Tillage*Fertilizer | 0.138 | 0.236 | 0.117 | 0.209 | 0.986 |
| (b) | | | | | |
| | **3-Mar** | **10-Mar** | **17-Mar** | **24-Mar** | **31-Mar** |
| Tillage | | | | | |
| Cf | 31.0 | 29.9 | 34.7 | 31.9 | 33.6 |
| F | 34.5 | 33.6 | 35.6 | 35.0 | 34.3 |
| R | 25.3 | 23.1 | 29.1 | 26.1 | 28.6 |
| LSD ($p < 0.05$) | 7.6 | 9.6 | 7.9 | 8.1 | 4.6 |
| Fertilizer | | | | | |
| None | 29.8 | 28.4 | 32.9 | 31.7 | 31.7 |
| Manure | 29.2 | 26.5 | 30.9 | 28.1 | 32.2 |
| Mineral | 32.0 | 31.7 | 35.7 | 33.2 | 32.7 |
| LSD ($p < 0.05$) | 8.4 | 10.4 | 8.2 | 8.6 | 5.2 |
| ANOVA | | | *p* value | | |
| Tillage | 0.016 * | 0.035 * | 0.136 | 0.018 * | 0.065 |
| Fertilizer | 0.59 | 0.379 | 0.364 | 0.215 | 0.926 |
| Tillage*Fertilizer | 0.132 | 0.112 | 0.488 | 0.062 | 0.666 |

Cf, conventional flat-bed tillage plot; F, furrow plot; R, ridge plot. *, **, and *** indicate significance at 0.05, 0.01, and 0.001 probability levels, respectively.

The fertilization effect of the SPAD value was significant at the 0.1% level in both measurements, and there was a significant interaction between tillage and fertilization treatment at the 1% level at 92 days after seeding (Table 6). Manure application increased the SPAD values, whereas mineral fertilization had the opposite effect. In furrow areas of the R/F treatment, manure fertilization increased SPAD values at the late maturity stage. Thus, the manure treatment (approximately 95 kg N/ha) was more effective than the mineral N treatment (30 kg N/ha) in maintaining the greenness of pearl millet into the seed setting stage.

**Table 6.** Averages and results of ANOVA for SPAD in 2018.

|  | 17-MAR (71 DAS) | 7-APR (92 DAS) |
|---|---|---|
| Tillage |  |  |
| CT | 60.9 | 48.5 |
| F | 62.8 | 48.5 |
| R | 63.3 | 50.5 |
| LSD ($p < 0.05$) | 2.3 | 4.1 |
| Fertilizer |  |  |
| None | 62.4 | 45.9 |
| Manure | 66.2 | 57.1 |
| Mineral | 58.6 | 44.5 |
| LSD ($p < 0.05$) | 2.8 | 5.6 |
| ANOVA | *p* value | |
| Tillage | 0.048 * | 0.433 |
| Fertilizer | 0.000 *** | 0.000 *** |
| Tillage*Fertilizer | 0.273 | 0.002 ** |

Cf, conventional flat-bed tillage plot; F, furrow plot; R, ridge plot; *, **, *** indicate significance at 0.05, 0.01, 0.001 probability levels, respectively.

## 4. Discussion

In the present study, the average yield of pearl millet was 0.73 t ha$^{-1}$ in 2017 and 2.96 t ha$^{-1}$ in 2018. Although the early to mid-growth periods in 2018 were too dry for regular pearl millet cultivation, the yield in the experimental field was higher than that in 2017. Consequently, in 2017, pearl millet was considerably damaged by waterlogging stress at the early to mid-growth stage, with the induction of hypoxia or anoxia resulting in a lower yield than in 2018. Thus, higher yields of pearl millet can be achieved in seasonal wetlands even in drought environments during the early to mid-growth periods, because of the suitable soil moisture content and fertility level for pearl millet growth [10]. In 2017, the 30 kg N/ha of mineral fertilizer was associated with an increase in yield, while the higher N rate of approximately 95 kg N/ha, as manure had a lower yield than mineral fertilization. This may have been because there was insufficient time for it to mineralize and become available to the crop. In 2018, there would have been more time for the manure N to mineralize, because there was also residual manure from the previous year. Thus, continuous manure administration might be important for improving crop production. The seasonal wetlands are predominantly characterized by sandy soils with low organic matter content in the study site; therefore, the small amount of rainfall is subject to high infiltration and evaporation rates. In fact, the water that escaped the surface through infiltration was concentrated in the deeper soil layers. The differences in tillage and fertilization effects in both years on the yield can be attributed to the different prevalent moisture conditions resulting from the rainfall pattern in each case.

Our results show that ridge-furrow tillage combined with manure application resulted in the highest yield in 2018. Thus, incorporating this method of cultivation could enhance crop growth and increase yield during the drought years in this region. Although the yield of pearl millet in ridges was higher than that in furrows throughout the 2-year experiment period (Table S1), the cultivation in furrows in seasonal wetlands might be an effective approach during considerable drought periods. Indeed, a lack of rainfall and low soil moisture content led to a severe and prolonged drought in 2019, and it was ranked as the most severe drought in the last 90 years [32]. The ridge-furrow system provides some protection for crop yields from flooding, because the ridge plants produce better in flooded years and furrow plants may produce better in drought years. Ridge-furrow tillage represents an effective way of distributing risk in areas where such extreme droughts and floods have become recurrent.

Pearl millet responded to an increased nutrient supply primarily by increasing the panicle density [33]. The panicle density was considered important for enhancing the yield in both years. Thus, applying additional fertilizers during the tillering stage and/or

continuous organic fertilization might be necessary to enhance crop production. Mineral fertilization promoted early plant growth, which allowed the plants to overcome soil moisture stress during the mid-growth period especially in 2017, when more flooding occurred. In comparison, the application of manure fertilizer led to a higher yield and HI in 2018. Thus, manure fertilization improved growth during the later stages in March 2018, whereas plots of the fertilizer treatment showed lodging during the heavy rainfall in the reproductive stage. In addition, manure fertilizer improved the water use efficiency, and the tendency agreed with a previous study [34]. The soils with improved carbon by manure fertilizer retained less water.

Soil OC is a key element for sustainable agriculture. Improved OC accumulation is associated with greater microbial and root growth, nutrient and water supply, soil aggregation, and better pH and temperature regulation [35]. In general, the OC content is extremely low in the agricultural fields of Namibia [15]. Appropriate manure application might improve both soil fertility and drainage. The amount of OC was not particularly affected by administering the manure fertilizer, probably because of the large amount of $CO_2$ emitted from the decomposition of OC in sandy land [36]. Khan et al. [37] reported that the OC content was 18% higher in soil after the application of dairy manure (20 Mg ha$^{-1}$ year$^{-1}$) than in the control. The amount of manure applied in this study (10 Mg ha$^{-1}$ year$^{-1}$) was considered to be insufficient to increase the OC content in sandy soils.

Plants in the unfertilized plots were considerably smaller than those in other plots, implying that they might not have absorbed sufficient nutrients, although the contents of N and K were the highest in these plots. Mineral fertilization plots showed the lowest SPAD values and soil N content during the seed setting stage. Mineral fertilizer enhanced the early growth of pearl millet; however, this phenomenon might cause lodging and nutrient deficiency later in the growth cycle should the crop encounter excess moisture when plants reach maturity. In comparison, manure fertilizer application resulted in the highest SPAD and pH values, and had the lowest soil moisture content during the late growth stage. In 2018, yield was closely related to the grain density, and manure fertilizer might enhance physiological traits and grain filling during the seed setting stage, and as a result increase yield. Thus, manure application improved soil conditions and, thereby, plant physiological activities during the late growth period. Conversely, pearl millet plants grown in the mineral fertilization plots were N-deficient during the late growth period, resulting in a lower yield. Soil N content was higher in the ridged plots, whereas P and K content was higher in the furrowed plots. In addition, the responsiveness of some yield components to fertilization methods was different between the ridges and furrows (Table S1). Thus, different methods of fertilization are required for ridges and furrows to improve soil quality and crop productivity.

An analysis of crop growth is important to monitor crops, predict crop yield, make field management recommendations, evaluate agricultural production potential, and assess the potential effects of climate change [38,39]. Hirooka et al. [40] underlined the importance of analyzing crop growth to elucidate how cultivation methods affect soil moisture content and fertility. In particular, the results of this study reveal that tillering is important to enhancing yield, and the evaluation of the effects of cultivation management on growth during the vegetative stage is needed. The unfertilized plot showed considerably lower grain density as well as panicle density in 2018, and fertilization is considered important in this area. In addition, the planting date is an important factor for improving crop productivity [41]. In seasonal wetlands, ridge-furrow tillage mitigated the risk of crop loss, and manure and/or additional fertilizer application at around tillering might enhance crop yield. The findings of this study warrant further research on crop growth to determine the optimum amount of fertilizer required and the timing of application.

## 5. Conclusions

The optimal cultivation management of pearl millet in seasonal wetlands differed with the rainfall patterns (2017 and 2018). While mineral fertilization enhanced the early growth

of pearl millet, it caused lodging and nutrient deficiency in the late growth stage in 2018. In comparison, continuous manure fertilization improved the growth of plants during the later stages. We concluded that mineral fertilizer is efficient for mitigating the waterlogging stress in the early to mid-growth stage, and manure fertilizer is efficient for mitigating the environmental stress in the late growth stage. Yield was significantly related to panicle density in both years, and therefore split applications of N or a slow-release source such as manure might be needed to enhance crop yield. Ridge-furrow tillage represents an effective way to distribute crop loss in these areas, and different fertilization methods for ridges and furrows were concluded to be necessary to improve soil quality. Although manure fertilization could reduce the risk and enhance crop yield if integrated with ridge-furrow tillage, the application of additional fertilizer based on the growth diagnostics could improve crop production in seasonal wetlands.

**Supplementary Materials:** The following are available online at https://www.mdpi.com/article/10.3390/agronomy11091767/s1, Table S1: Means for comparison of yield and yield components of pearl millet between ridge and furrow plots.

**Author Contributions:** Conceptualization, Y.H. and S.K.A.; methodology, S.K.A.; validation, S.K.A., K.H. and P.I.N.; formal analysis, Y.H. and K.H.; investigation, S.A and P.I.N.; resources, S.K.A.; data curation, S.K.A. and P.I.N.; writing—original draft preparation, Y.H.; writing—review and editing, S.K.A., K.H., P.I.N. and M.I.; visualization, Y.H.; supervision, M.I.; project administration, M.I.; funding acquisition, Y.H. All authors have read and agreed to the published version of the manuscript.

**Funding:** This work was supported by the project entitled 'Flood- and Drought-adaptive Cropping Systems to Conserve Water Environments in Semi-arid Regions' by the framework of the 'SATREPS' funded by both the Japan Science and Technology Agency (JST) and Japan International Cooperation Agency (JICA). This work was also supported by Application Procedures for Fund for the Promotion of Joint International Research [grant number 19KK0158], Grant-in-Aid for Research Activity start-up [grant number 16H07355] and Grant-in-Aid for Young Scientists [grant number 18K14454].

**Data Availability Statement:** The data presented in this study are available on request from the corresponding author.

**Acknowledgments:** We thank technical staff and students of the University of Namibia for cultivation management during the crop cycles comprised in this study.

**Conflicts of Interest:** The authors declare no conflict of interest.

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
