# Peer review of "Effects of Cultivation Management on Pearl Millet Yield and Growth Differed with Rainfall Conditions in a Seasonal Wetland of Sub-Saharan Africa"

_agronomy, doi:10.3390/agronomy11091767_

Round 1
Reviewer 1 Report
Dear authors
Your paper is very carefully and professionally written, in excellent English! There are, however, several major weaknesses of which only the second can be corrected at this stage:
(1) There is no "concept" behind this paper, no logical framework as those provided by crop models. In this sense, your paper very much reminds me of the old purely descriptive agronomic literature of the late 20th century; (2) there exists a huge volume of pearl millet and ridging research in semi-arid areas, of which some overview needs to be be provided and (3) a point related to (1), i.e. the paper is very weak on environmental determinants of yield, in particular weather which is dealt with very vaguely. For instance, your data show that there are some water use efficiency issues, which are nor addressed.
The list below provides additional comments, including some minor points about language, on a line-by-line basis. Note that when I conventionally write "L 427 Africa → Asia" I suggest that "Africa" should become "Asia" on line 427.
L 26, 52, 271, 272, 279, 282, 318, 324, 332, 338: Productivity.
Comment: productivity is an economic concept that is not well defined in an agronomic context. Please use “yield” or “production”, according to context
L 33-34: Environmental conditions in these areas are marginal, leading to land degradation → Human activity in these areas leads to land degradation
Comment: environmental conditions are what they are. They are not “good” or “bad” nor “marginal”. It is human activities that lead to the listed problems.
L 34-35: Sub-Saharan Africa → Semi-arid Sub-Saharan Africa
Comment: Many areas in Sub-Saharan Africa enjoy abundant and regular precipitation
L 40-41: Recurring extreme drought and flood conditions → Extreme drought and floods
Comment: since “frequent” (the word is used later in the sentence) and “recurring” are the same concept, it does not need repeating! The word “conditions” is unnecessary
L 49: provide an effective and → improve
L 51: strongly recommended
Comment: recommendations do not belong in the Introduction
L 55-56: Ridge-furrow tillage represents an efficient approach for the crops to be well adapted to these contrasting and unpredictable environments in these regions. → Ridge-furrow tillage represents an efficient approach in these contrasting and unpredictable environments.
L 57: creates a differential soil-moisture environment → modifies the soil-moisture environment
Comment: the use of “differential” is improper
L 69: interaction effects → interactions
L 69-73:
Comment: The authors have not been doing their homework. There is a huge volume of literature of ridging for millet(s) and other crops (e.g. sweet potatoes) in semi-arid climates in Africa and elsewhere (Sahel, East Africa, India, China and semi-arid Asia, the Central American Drought corridor etc). This paper is NOT acceptable if authors do not show that they are aware of this research, which has been going on for the last 50 years across the globe. The available literature also shows that ridging is a management technique among many others, including optimisation of planting dates, varietal selection, mixed and inter-cropping cropping.
L 83: Distribution of rainfall in the experimental region is shown in Figure 1 → The recent time profile of rainfall in the experimental region is shown in Figure 1
Comment: there’s a lot to say about this figure: (1) is this Ogongo rainfall or some “regional average”, which is hardly relevant given the spatial variability of rainfall in semi-arid areas? (2) is this Jan-Dec rainfall – which includes two cropping seasons - or some, more meaningful, rainy season rainfall (e.g. Aug-Jul)? The listed SASSCAL website has Ogongo rainfall only from 2013. So what is this profile in figure 1? See attached figure.
L 123: GW, PH, PL, TDW etc
Comment: Mention that units are listed in the Tables (e.g. Table 2). How is your HI defined: ratio of grain in above-ground biomass or total biomass?
L 203: Table 3
Comment. It is not clear to me which data have been used in this table. You have 3 cultivation typologies (F/R and CF), 3 fertilizer levels (none, manure, mineral), 2 years (2017 and 2018) and 3 replicates, which is 3 x 3 x 3 x 3 = 81 observations. Please mention the number of observations in the legend or the footnote to the tables.
L 227: The furrow plots presented significantly higher soil moisture content
Comment: are the furrows oriented perpendicularly or parallel to the slope?
L 228: trend → pattern
L 229: Fertilization had no significant effect on the soil moisture content.
L 230-231: However, the manure-fertilized plots tended to have lower soil moisture content than other plots.
Comment: this is very strange. According to table 2, fertilizer boosted biomass accumulation. What you are saying, is that water use efficiency also varied among your plots. On the other hand, the soils with improved carbon (table 4) retain less water. This deserves some comments!
L 282-297
Comment: this is close to random speculation. If you had a third year of observations, you would have completely different conclusions.

Author Response
Dear Reviewer 1:
Your comments: Your paper is very carefully and professionally written, in excellent English! There are, however, several major weaknesses of which only the second can be corrected at this stage:
Our answer: Thank you for the useful and careful reviewing. We thoroughly revised our manuscript according to your comments.
Your comments: (1) There is no "concept" behind this paper, no logical framework as those provided by crop models. In this sense, your paper very much reminds me of the old purely descriptive agronomic literature of the late 20th century; (2) there exists a huge volume of pearl millet and ridging research in semi-arid areas, of which some overview needs to be provided and (3) a point related to (1), i.e. the paper is very weak on environmental determinants of yield, in particular weather which is dealt with very vaguely. For instance, your data show that there are some water use efficiency issues, which are nor addressed.
Our answer: It is sure that there is not modelling approach in this study, but the manuscript is about the potential advantage of cultivation management which could improve the pearl millet production in Namibia in SSA region. We are planning to conduct modelling approach after getting further data (at least 5 years). We added the previous study about pearl millet and ridging research in semi-arid areas in Introduction part and discussed the water use efficiency in Discussion part, according to your comments.
Your comments: The list below provides additional comments, including some minor points about language, on a line-by-line basis. Note that when I conventionally write "L 427 Africa → Asia" I suggest that "Africa" should become "Asia" on line 427.
Our answer: Thank you very much. We revised our manuscript according to your comments.
Your comments: L 26, 52, 271, 272, 279, 282, 318, 324, 332, 338: Productivity.
Comment: productivity is an economic concept that is not well defined in an agronomic context. Please use “yield” or “production”, according to context
Our answer: We revised “productivity” to “production” or “yield”.
Your comments: L 33-34: Environmental conditions in these areas are marginal, leading to land degradation → Human activity in these areas leads to land degradation
Comment: environmental conditions are what they are. They are not “good” or “bad” nor “marginal”. It is human activities that lead to the listed problems.
Our answer: We revised it according to your comments (Line 34-35).
Your comments: L 34-35: Sub-Saharan Africa → Semi-arid Sub-Saharan Africa
Comment: Many areas in Sub-Saharan Africa enjoy abundant and regular precipitation
Our answer: We added the phrase (Line 36).
Your comments: L 40-41: Recurring extreme drought and flood conditions → Extreme drought and floods
Comment: since “frequent” (the word is used later in the sentence) and “recurring” are the same concept, it does not need repeating! The word “conditions” is unnecessary
Our answer: We revised the sentence (Line 42-43).
Your comments: L 49: provide an effective and → improve
Our answer: We revised it (Line 51).
Your comments: L 51: strongly recommended
Comment: recommendations do not belong in the Introduction
Our answer: We removed the sentence (Line 53-54).
Your comments: L 55-56: Ridge-furrow tillage represents an efficient approach for the crops to be well adapted to these contrasting and unpredictable environments in these regions. → Ridge-furrow tillage represents an efficient approach in these contrasting and unpredictable environments.
Our answer: We revised the sentence (Line 57-58).
Your comments: L 57: creates a differential soil-moisture environment → modifies the soil-moisture environment
Comment: the use of “differential” is improper
Our answer: We revised the phrase (Line 59).
Your comments: L 69: interaction effects → interactions
Our answer: We revised it (Line 74).
Your comments: L 69-73:
Comment: The authors have not been doing their homework. There is a huge volume of literature of ridging for millet(s) and other crops (e.g. sweet potatoes) in semi-arid climates in Africa and elsewhere (Sahel, East Africa, India, China and semi-arid Asia, the Central American Drought corridor etc). This paper is NOT acceptable if authors do not show that they are aware of this research, which has been going on for the last 50 years across the globe. The available literature also shows that ridging is a management technique among many others, including optimisation of planting dates, varietal selection, mixed and inter-cropping cropping.
Our answer: As mentioned above, we show that there are many previous studies about ridging for millets and combination of riding with cultivation management (Line 71-74).
Your comments: L 83: Distribution of rainfall in the experimental region is shown in Figure 1 → The recent time profile of rainfall in the experimental region is shown in Figure 1
Comment: there’s a lot to say about this figure: (1) is this Ogongo rainfall or some “regional average”, which is hardly relevant given the spatial variability of rainfall in semi-arid areas? (2) is this Jan-Dec rainfall – which includes two cropping seasons - or some, more meaningful, rainy season rainfall (e.g. Aug-Jul)? The listed SASSCAL website has Ogongo rainfall only from 2013. So what is this profile in figure 1? See attached figure.
Our answer: We revised the part according to your comments (Line 88). The rainfall data was measured at Omahenene weather station (17°27′S, 14°47′E) near Ogongo. We re-made the Figure (rainy season rainfall (Aug-July)), and added the information above (Line 89, 106).
Your comments: L 123: GW, PH, PL, TDW etc
Comment: Mention that units are listed in the Tables (e.g. Table 2). How is your HI defined: ratio of grain in above-ground biomass or total biomass?
Our answer: We removed the acronyms to interpret easily according to the Reviewer 2 comments. Harvest index (HI) was calculated as the ratio of grain yield and total dry weight at maturity (Yield/TDW) (Line 151).
Your comments: L 203: Table 3
Comment. It is not clear to me which data have been used in this table. You have 3 cultivation typologies (F/R and CF), 3 fertilizer levels (none, manure, mineral), 2 years (2017 and 2018) and 3 replicates, which is 3 x 3 x 3 x 3 = 81 observations. Please mention the number of observations in the legend or the footnote to the tables.
Our answer: The correlation coefficient was calculated by the cultivation typologies (2 levels), the fertilizer application (3 levels), and 3 replicates (2 x 3 x 3 = 18 observations) in 2017 and 2018, respectively. We added the number of observations in the footnote of Table 3.
Your comments: L 227: The furrow plots presented significantly higher soil moisture content
Comment: are the furrows oriented perpendicularly or parallel to the slope?
Our answer: The furrows were oriented perpendicularly. However, the experimental field (lower part of the slope field) had not so much inclination.
Your comments: L 228: trend → pattern
Our answer: We revised the word (Line 268).
Your comments: L 229: Fertilization had no significant effect on the soil moisture content.
L 230-231: However, the manure-fertilized plots tended to have lower soil moisture content than other plots.
Comment: this is very strange. According to table 2, fertilizer boosted biomass accumulation. What you are saying, is that water use efficiency also varied among your plots. On the other hand, the soils with improved carbon (table 4) retain less water. This deserves some comments!
Our answer: We could not conclude which the treatment improved water use efficiency because we have only 2 year and 1 site data. However, manure fertilizer treatment might improve water use efficiency as you said. We added the discussion with previous study (Line 343-346).
Your comments: L 282-297
Comment: this is close to random speculation. If you had a third year of observations, you would have completely different conclusions.
Our answer: We concluded that mineral fertilizer is efficient for mitigating the waterlogging stress in the early growth stage, and manure fertilizer is efficient for mitigating the environmental stress in the late growth stage. So, if the third year is conducted, the result might be depended on the rainfall pattern (the stress at the early growth stage or late growth stage). We added the explanation in conclusion part (Line 397-399).
Reviewer 2 Report
This paper describes a study that appears to have been well conducted. The Abstract reads well. However, the conclusions drawn are inconsistent with the evidence presented. Firstly, the comparison between chemical fertiliser and manure is flawed by large differences in rates. The synthetic fertiliser was applied at only 30 kg N/ha at sowing, which is a low rate relative to N uptake into the plant. This is compared with manure applied at 10 t/ha, which at about 2% N is 200 kg N/ha. Wording should be changed to indicate that this is a low rate of synthetic N, and an estimate provided of the N contained and other nutrients in the manure. Some context should be provided as to how realistic these rates are in commercial practice. For example, 30 kg N/ha may be typical of what local farmers apply. How realistic is the 10 t/ha of manure? For a rangeland growing 5 t/ha of herbage grazed by cattle at a utilisation rate of 50% and with a herbage digestibility of 50%, this is a manure production rate of only 1.25 t/ha. A manure application rate of 10 t/ha on the crop this would require that for each hectare of crop either manure be collected from 8 hectares, or herbage collected from 8 hectares and stall-fed.
Secondly a conclusion is made that manure did not affect soil carbon. A manure application rate of 10 t/ha with a 40% carbon content and a topsoil (0-10 cm) bulk density of 1.0 should increase the soil carbon content by 0.4% (40 g/kg) less any organic carbon respired over the year. A second year of application should increase it by the same amount again. In Table 4 the organic carbon content, presumably of the 0-10 layer, is quoted as 2.44 g/kg for the unfertilised treatment and 2.96 for manured, implying that nearly all the added 80 g/kg has disappeared. These organic carbon contents are extremely low for tropical soils, and I expect the units should instead be %. So the difference is instead 2.96% for manured minus 2.44% for control or 0.52%, which is consistent with the amount of manure added (0.8%) less any that has decomposed.
The Results section is too detailed. An example of rewording is shown below. Each Table only needs to be referred to in one or two sentences, highlighting the points that are later taken up in the Discussion. The Tables are unusual in their presentation of sums of square. These values are of no use to readers. Instead the 5% lsd should be presented and P values for the probability of an effect.
There is also excessive use of brackets.
Specific rewording suggestions
Lines 25-26 Remove mention of long-range weather forecasts as no data are presented on this
Line 41 “semi-arid northern regions of the country”
Line 45-46 “ wetlands, which are the lower sections of field slopes influence the yield”
Line 48-49 “Moreover, cultivation of seasonal wetlands could provide an effective agricultural production system in northern Namibia” [this study doesn not address sustainability so there is no need for it in the Introduction]
Line 55 “for crops adapted…environments.”
Line 73 “cultivation under highly variable rainfall conditions”
Figure 1. Are these values for a single station, in which case the location name, latitude and longitude should be given. If it is an area average calculated from several stations a citeable published data source should be quoted.
Lines 104-114 An estimate of NPK application in the manure should be provided. This should be based on manure measurements from animals fed a diet similar to those in the experiment. For example, dairy cows are normally fed a much richer diet than rangeland beef cattle, so their manure would be richer in N. Or was an the manure analysed so these calculations could be made specially for this manure?
At what depth was the fertiliser applied ? Was it surface-applied or with the seed?
At what depth was the seed sown?
Was the manure surface applied or worked into the soil?
Was the manure applied once-only in the first year or a second application applied prior to the second year crop?
Was the synthetic fertiliser applied again in the second year?
How long was the interval between manure application and sowing? How much rain fell over this period (which could cause mineralisation over this period)?
Were the same plot treatment allocations used in second year as in the first?
How was the land managed between the first and second year crops?
Lines 154-159 Were there any signs of surface water movement onto the plots or short term flooding? This is central to the hypothesis but no information is provided. Ideally measurements of this should have been made, such as mm of run-on and days and depth of flooding. If such measurements were not taken, a summary of diary notes would be insightful, such as “signs of surface water movement and short term (<3 days) flooding of the crop were observed during grain-fill following heavy rainfall in March 2017, but no flooding was observed in 2018”.
Line 157 Delete “lower part of the slope” as this is in the Methods
Table 1 Use an extra 5 rows for 2018. Use of brackets in this way is unusual and makes the Table more difficult to interpret.
Lines 162-176 There are too many acronyms in this paper. Most readers can only remember 5 for a paper, so choose only those variables that are used most frequently, or used commonly in other papers (such as HI for harvest index). Others such as PN should be used in full, because in other papers it could represent “plant number” but here is panicle density.
Results presentation is too long, and should instead only highlight the significant points from each Table. Here is a suggested rewording of lines 162-178: “Grain yield in 2017 ranged from an average of 0.49 t/ha on the unfertilised treatment to 1.02 t/ha where mineral fertiliser had been added (Table 2). In 2018 the range in grain yields was from 2.0 t/ha on the unfertilised treatment to 3.97 t/ha for the manured treatment. In neither year was there a response in grain yield to tillage treatment. Harvest indices averaged 0.22 in 2017 and 0.44 in 2018, and were not significantly affected by either tillage nor fertiliser treatment. The higher yield on fertilised treatments was through a higher density of panicles, and more grains per panicle (in 2018). Grain weight was lower in the R/T treatment in 2017, and was not significantly affected by fertiliser treatment. Fertiliser was associated with an increase in plant height in 2017, while panicle length was unaffected by either treatment. Grain yield was most strongly correlated with maturity biomass, followed by panicle density (Table 3).”
Some aspects of this rewriting style may be worthwhile incorporating into other Results paragraphs.
Table 2. Use headings split over two lines “Grain yield”, “Maturity biomass”, “Harvest index”, “Panicle density”, ‘Grain density”, “Grain weight”, Plant height”, “Panicle length”
The years should be centred so more obvious, eg
------------------------2017-----------------------
Mean squares do not need to be presented. Instead for each year present the 5% lsd and P value for tillage and fertiliser effects in each year. P values should be presented as continuous values (eg 0.898) rather than discrete (eg ns, <0.05) as this provides information on strength of the effects. Each year should be analysed separately. Year effects were significant for all but one parameter, but it’s not useful because we know the years were different. What is more important is whether the tillage and fertiliser effects were significantly different, and the consistency of differences across the years. Other Tables should be presented in this way as well.
The footnote says the letters separate significant differences, but there are no such letters in the Table despite there being some significant differences.
Use “Tillage and Fertiliser” rather than “T” and “F” as it’s quicker for the reader to understand.
Lines 183 to 199 This is boring to read and much better presented in Table form. Some of the suggested rewording above should address this.
Lines 215 to 216 Delete “Plants in the…these plots” as this is interpretation that should be moved into Discussion.
Table 4 Title should be “Residual effects of tillage and fertiliser treatments on soil characteristics (0-10 cm) in April 2018 after completion of the second year of the experiment.
Table 5. Gravimetric soil moisture (g water/g dry soil) at depths of (a) 10-20 cm and (b) 30-40 cm during March 2018.
Line 245 “In furrow areas of the R/F treatment…”
Why was there no separate analysis of plants on the ridge vs furrow, as there was for soil samples?
One of the points made in discussion is that the ridge/furrow system provides some protection of yields from flooding, because the ridge plants produce better in flooded years and furrow plants may produce better in minimally flooded years.
Line 246-249 “Thus…nutritional deficiency” Can’t see support for this in the data. Better to conclude “Thus, the manure treatment (200 kg N/ha) was more effective than the mineral N treatment (30 kg N/ha) at maintaining crop greenness into the grain-fill period.”
Table 6 Add the number of days after sowing (71 and 92 days) within each column.
Line 255-256 “0.75 t/ha in 2017 and 2.96 t/ha in 2018” Try to avoid use of the term “respectively”
Line 255-268 This section is not strong because there has not been a strong description of the two seasons at the start of the Results section. This first paragraph should draw together the author’s understanding of the two seasons. In 2017 the 30 kg N/ha of mineral fertiliser was associated with an increase in yield over the nil control. However, the higher N rate of 200 kg N/ha as manure had a lower yield. This may have been because there had been insufficient time for it to mineralise and become available to the crop, or because it caused the soil to become anoxic during flooding in the grainfill period. In 2018 there would have been more time for the manure N to mineralise, because there was also residual manure from the previous year. Also, fewer complications due to waterlogging during the grainfill period.
Were the tops of the ridges flooded, and for how long?
Did any flooding occur in 2018?
How did plants on the ridges compare with plants in the furrows?
Observations of growth stage would help, because most plants are more vulnerable to waterlogging during grainfill.
The first two paragraphs of Discussion need to be completely rewritten into a single strong statement about the author’s interpretation of the two seasons affected how the crop responded to the tillage and fertility treatments.
Line 279 Pearl millet responded to an increased nutritent supply primarily by increasing the panicle density.
Line 284 “especially in 2017, when more flooding occurred.”
Line 286 “whereas plots of the mineral fertiliser treatment showed”
Line 291 “sufficient OC content” This needs a reference relating OC to crop growth otherwise omit. It would be worthwhile citing a range of OC in agricultural fields from reference [25], since it is unlikely to be available to most readers.
Line 296 Wasn’t 20 Mg/ha of organic matter added? It is unclear in the Methods whether the manure was added once in 2017 or a second application in 2018.
This section needs to explain by how much the manure application would be expected to increase OC and compare it to the 5% lsd. The OC may have increased but by less than the statistical detection limit.
Line 301 “cycle should the crop encounter excess”
Line 302 “values, and had the”
Line 332 “therefore split applications of N or a slow-release source such as manure might”
Line 334 “furrows were concluded to be” The logic behind this conclusion is not in the Discussion. It would be a useful Discussion point and conclusion
Author Response
Dear Reviewer 2
Your comments: This paper describes a study that appears to have been well conducted. The Abstract reads well. However, the conclusions drawn are inconsistent with the evidence presented.
Our answer: Thank you for the useful and careful reviewing. We revised our manuscript thoroughly according to your comments.
Your comments: Firstly, the comparison between chemical fertiliser and manure is flawed by large differences in rates. The synthetic fertiliser was applied at only 30 kg N/ha at sowing, which is a low rate relative to N uptake into the plant. This is compared with manure applied at 10 t/ha, which at about 2% N is 200 kg N/ha. Wording should be changed to indicate that this is a low rate of synthetic N, and an estimate provided of the N contained and other nutrients in the manure. Some context should be provided as to how realistic these rates are in commercial practice. For example, 30 kg N/ha may be typical of what local farmers apply. How realistic is the 10 t/ha of manure? For a rangeland growing 5 t/ha of herbage grazed by cattle at a utilisation rate of 50% and with a herbage digestibility of 50%, this is a manure production rate of only 1.25 t/ha. A manure application rate of 10 t/ha on the crop this would require that for each hectare of crop either manure be collected from 8 hectares, or herbage collected from 8 hectares and stall-fed.
Our answer: It is sure that the amount of N based fertilizer was different between manure and mineral. However, we set the amount of manure and mineral fertilizer based on the local recommendation in the target area. We added the information (Line 117-118). Of course, further studies of the different amount of manure and mineral fertilizer were required.
Your comments: Secondly a conclusion is made that manure did not affect soil carbon. A manure application rate of 10 t/ha with a 40% carbon content and a topsoil (0-10 cm) bulk density of 1.0 should increase the soil carbon content by 0.4% (40 g/kg) less any organic carbon respired over the year. A second year of application should increase it by the same amount again. In Table 4 the organic carbon content, presumably of the 0-10 layer, is quoted as 2.44 g/kg for the unfertilised treatment and 2.96 for manured, implying that nearly all the added 80 g/kg has disappeared. These organic carbon contents are extremely low for tropical soils, and I expect the units should instead be %. So the difference is instead 2.96% for manured minus 2.44% for control or 0.52%, which is consistent with the amount of manure added (0.8%) less any that has decomposed.
Our answer: The unit (g/kg) is correct. The total C in the study area was very low. As you said, the total C should be increased in manure fertilized plot. However, the organic carbon was only 0.52g/kg increasing. The organic carbon was not increased in the sandy soil more than expected. The tendency was similar to Watanabe et al. (2019), which reported that the organic carbon was not increased by manure fertilizer so much in the sandy soil. This is probably because of the large amount of CO2 emission from the decomposition of OC in sandy land (Zhenghu et al., 2001). We revised the paragraph of organic carbon in Discussion part (Line 347-359).
Your comments: The Results section is too detailed. An example of rewording is shown below. Each Table only needs to be referred to in one or two sentences, highlighting the points that are later taken up in the Discussion. The Tables are unusual in their presentation of sums of square. These values are of no use to readers. Instead the 5% lsd should be presented and P values for the probability of an effect.
Our answer: Thank you for your kindful revising the Result part. We revised the Result section, based on the revising. In addition, we removed the sum of square, and added 5%LSD and P value in Table 2, 4, 5, and 6.
Your comments: There is also excessive use of brackets.
Our answer: As mentioned below, we removed the brackets.
Your comments: Lines 25-26 Remove mention of long-range weather forecasts as no data are presented on this
Our answer: We removed the word in Abstract and Conclusion (Line 26, 406-407).
Your comments: Line 41 “semi-arid northern regions of the country”
Our answer: We revised the part (Line 43-44).
Your comments: Line 45-46 “ wetlands, which are the lower sections of field slopes influence the yield”
Our answer: We revised the part (Line 47-48).
Your comments: Line 48-49 “Moreover, cultivation of seasonal wetlands could provide an effective agricultural production system in northern Namibia” [this study doesn not address sustainability so there is no need for it in the Introduction]
Our answer: We revised the part according to your comment and Reviewer 1 comment (Line 51-52).
Your comments: Line 55 “for crops adapted…environments.”
Our answer: We revised the sentence according to Reviewer 1 comment (Line 57-58).
Your comments: Line 73 “cultivation under highly variable rainfall conditions”
Our answer: We revised the part (Line 77-78).
Your comments: Figure 1. Are these values for a single station, in which case the location name, latitude and longitude should be given. If it is an area average calculated from several stations a citeable published data source should be quoted.
Our answer: The rainfall data was measured at the weather station in Omahenene (17°27′S, 14°47′E) in the target area, northern Namibia. We added the information (Line 88, 106).
Your comments: Lines 104-114 An estimate of NPK application in the manure should be provided. This should be based on manure measurements from animals fed a diet similar to those in the experiment. For example, dairy cows are normally fed a much richer diet than rangeland beef cattle, so their manure would be richer in N. Or was an the manure analysed so these calculations could be made specially for this manure?
Our answer: Watanabe et al. (2019) reported that almost half of the amount of the cattle manure was organic matter, and the total N, P, and K were 9.5, 1.6, and 4.7 g kg−1, respectively. We added the information (Line 122-123).
Your comments: At what depth was the fertiliser applied ? Was it surface-applied or with the seed?
Our answer: Manure fertilizer was surface applied based on the common practice by farmers in the target area. We added the information (Line 119-120).
Your comments: At what depth was the seed sown?
Our answer: The seed was sown at 3 cm depth. We added the information (Line 110).
Your comments: Was the manure surface applied or worked into the soil?
Our answer: Mineral fertilizer was surface applied based on common practice by farmers in the target area. We added the information (Line 119-120).
Your comments: Was the manure applied once-only in the first year or a second application applied prior to the second year crop? Was the synthetic fertiliser applied again in the second year?
Our answer: Both the manure and mineral fertilizer was applied twice before starting 2017 and 2018 experiments. We added the information (Line 120).
Your comments: How long was the interval between manure application and sowing? How much rain fell over this period (which could cause mineralisation over this period)?
Our answer: Manure fertilizer was applied four weeks before sowing. We added the information (Line 121). The rainfall during the period was 28.3 mm in 2017 and 74.4 mm in 2018.
Your comments: Were the same plot treatment allocations used in second year as in the first?
Our answer: Yes, the same plot treatment allocations were used in second year as in the first. We added the information (Line 118-119).
Your comments: How was the land managed between the first and second year crops?
Our answer: The land was managed using a rotary plow in Cf treatment and furrows, by hand hoeing in ridge between the first and second year crops. We added the information (Line 139-140).
Your comments: Lines 154-159 Were there any signs of surface water movement onto the plots or short term flooding? This is central to the hypothesis but no information is provided. Ideally measurements of this should have been made, such as mm of run-on and days and depth of flooding. If such measurements were not taken, a summary of diary notes would be insightful, such as “signs of surface water movement and short term (<3 days) flooding of the crop were observed during grain-fill following heavy rainfall in March 2017, but no flooding was observed in 2018”.
Our answer: Signs of surface water movement and short term flooding of the crop were observed in the early March during the reproductive growth stage in 2017 and early April during the seed setting stage in 2018. We added the information (Line 181-183).
Your comments: Line 157 Delete “lower part of the slope” as this is in the Methods
Our answer: We deleted the word (Line 179).
Your comments: Table 1 Use an extra 5 rows for 2018. Use of brackets in this way is unusual and makes the Table more difficult to interpret.
Our answer: We revised the Table 1 to interpret easily according to your comments.
Your comments: Lines 162-176 There are too many acronyms in this paper. Most readers can only remember 5 for a paper, so choose only those variables that are used most frequently, or used commonly in other papers (such as HI for harvest index). Others such as PN should be used in full, because in other papers it could represent “plant number” but here is panicle density.
Our answer: We removed PN, GN, GW, PH, and PL, and used them in full throughout the manuscript.
Your comments: Results presentation is too long, and should instead only highlight the significant points from each Table. Here is a suggested rewording of lines 162-178: “Grain yield in 2017 ranged from an average of 0.49 t/ha on the unfertilised treatment to 1.02 t/ha where mineral fertiliser had been added (Table 2). In 2018 the range in grain yields was from 2.0 t/ha on the unfertilised treatment to 3.97 t/ha for the manured treatment. In neither year was there a response in grain yield to tillage treatment. Harvest indices averaged 0.22 in 2017 and 0.44 in 2018, and were not significantly affected by either tillage nor fertiliser treatment. The higher yield on fertilised treatments was through a higher density of panicles, and more grains per panicle (in 2018). Grain weight was lower in the R/T treatment in 2017, and was not significantly affected by fertiliser treatment. Fertiliser was associated with an increase in plant height in 2017, while panicle length was unaffected by either treatment. Grain yield was most strongly correlated with maturity biomass, followed by panicle density (Table 3).” Some aspects of this rewriting style may be worthwhile incorporating into other Results paragraphs.
Our answer: Thank you for the careful revising Result part. We revised the paragraph according to your comments (Line 188-212), and revised other paragraphs based on your rewriting.
Your comments: Table 2. Use headings split over two lines “Grain yield”, “Maturity biomass”, “Harvest index”, “Panicle density”, ‘Grain density”, “Grain weight”, Plant height”, “Panicle length”
Our answer: We revised the headings of the Table 2 according to your comments.
Your comments: The years should be centred so more obvious, eg
------------------------2017-----------------------
Our answer: We devided Table 2 into two tables as mentioned below.
Your comments: Mean squares do not need to be presented. Instead for each year present the 5% lsd and P value for tillage and fertiliser effects in each year. P values should be presented as continuous values (eg 0.898) rather than discrete (eg ns, <0.05) as this provides information on strength of the effects. Each year should be analysed separately. Year effects were significant for all but one parameter, but it’s not useful because we know the years were different. What is more important is whether the tillage and fertiliser effects were significantly different, and the consistency of differences across the years. Other Tables should be presented in this way as well.
Our answer: According to your comments, we revised Table 2 for each year ((a) 2017, (b) 2018). We removed mean squares, and added the 5% lsd and P value presented as continuous values in Table 2, 4, 5, and 6.
Your comments: The footnote says the letters separate significant differences, but there are no such letters in the Table despite there being some significant differences.
Our answer: We removed the letters. Instead, we added the 5%LSD.
Your comments: Use “Tillage and Fertiliser” rather than “T” and “F” as it’s quicker for the reader to understand.
Our answer: We revised them in Table 2, 4, 5, and 6.
Your comments: Lines 183 to 199 This is boring to read and much better presented in Table form. Some of the suggested rewording above should address this.
Our answer: We revised the Result part thoroughly, based on your rewriting style.
Your comments: Lines 215 to 216 Delete “Plants in the…these plots” as this is interpretation that should be moved into Discussion.
Our answer: We moved the sentence to Discussion part (Line 360-362).
Your comments: Table 4 Title should be “Residual effects of tillage and fertiliser treatments on soil characteristics (0-10 cm) in April 2018 after completion of the second year of the experiment.
Our answer: We revised the title of Table 4, according to your comments (Line 260-261).
Your comments: Table 5. Gravimetric soil moisture (g water/g dry soil) at depths of (a) 10-20 cm and (b) 30-40 cm during March 2018.
Our answer: We revised the title of Table 5 (Line 273-275).
Your comments: Line 245 “In furrow areas of the R/F treatment…”
Our answer: We revised the part (Line 287).
Your comments: Why was there no separate analysis of plants on the ridge vs furrow, as there was for soil samples?
Our answer: In this study, we would like to compare yield and yield components of the conventional flatbed tillage with Ridge-furrow tillage. We added the ridge vs furrow yield and yield components as supplemental Table.
Your comments: One of the points made in discussion is that the ridge/furrow system provides some protection of yields from flooding, because the ridge plants produce better in flooded years and furrow plants may produce better in minimally flooded years.
Our answer: Thank you for suggesting the good point. We added the discussion in discussion part (Line 329-331).
Your comments: Line 246-249 “Thus…nutritional deficiency” Can’t see support for this in the data. Better to conclude “Thus, the manure treatment (200 kg N/ha) was more effective than the mineral N treatment (30 kg N/ha) at maintaining crop greenness into the grain-fill period.”
Our answer: We revised the part according to your comments (Line 288-290).
Your comments: Table 6 Add the number of days after sowing (71 and 92 days) within each column.
Our answer: We added the number of days after sowing in Table 6.
Your comments: Line 255-256 “0.75 t/ha in 2017 and 2.96 t/ha in 2018” Try to avoid use of the term “respectively”
Our answer: We removed the term and revised the part (Line 300-301).
Your comments: Line 255-268 This section is not strong because there has not been a strong description of the two seasons at the start of the Results section. This first paragraph should draw together the author’s understanding of the two seasons. In 2017 the 30 kg N/ha of mineral fertiliser was associated with an increase in yield over the nil control. However, the higher N rate of 200 kg N/ha as manure had a lower yield. This may have been because there had been insufficient time for it to mineralise and become available to the crop, or because it caused the soil to become anoxic during flooding in the grainfill period. In 2018 there would have been more time for the manure N to mineralise, because there was also residual manure from the previous year. Also, fewer complications due to waterlogging during the grainfill period.
Our answer: Thank you for your valuable comments. We added the explanation in the first paragraph in Discussion (Line 308-314).
Your comments: Were the tops of the ridges flooded, and for how long?
Our answer: The tops of the ridges was not flooded, so we revised the word “flooding stress” to “waterlogging stress” throughout the manuscript.
Your comments: Did any flooding occur in 2018?
Our answer: Signs of surface water movement and short term flooding of the crop were observed in the early April during the seed setting stage in 2018.
Your comments: How did plants on the ridges compare with plants in the furrows?
Our answer: As mentioned in Line 146, each 12 plants were compared.
Your comments: Observations of growth stage would help, because most plants are more vulnerable to waterlogging during grainfill.
Our answer: We added the information of growth stage during the flooding in 2017 and 2018 (Line 182-183).
Your comments: The first two paragraphs of Discussion need to be completely rewritten into a single strong statement about the author’s interpretation of the two seasons affected how the crop responded to the tillage and fertility treatments.
Our answer: As mentioned above, we revised the first two paragraphs of Discussion.
Your comments: Line 279 Pearl millet responded to an increased nutritent supply primarily by increasing the panicle density.
Our answer: We revised the sentence according to your comments (Line 333-334).
Your comments: Line 284 “especially in 2017, when more flooding occurred.”
Our answer: We revised the part (Line 339-340).
Your comments: Line 286 “whereas plots of the mineral fertiliser treatment showed”
Our answer: We revised the part (Line 342-343).
Your comments: Line 291 “sufficient OC content” This needs a reference relating OC to crop growth otherwise omit. It would be worthwhile citing a range of OC in agricultural fields from reference [25], since it is unlikely to be available to most readers.
Our answer: We removed the part, and revised the paragraph in Discussion (Line 347-359).
Your comments: Line 296 Wasn’t 20 Mg/ha of organic matter added? It is unclear in the Methods whether the manure was added once in 2017 or a second application in 2018.
Our answer: The manure fertilizer (10 Mg/ha) was applied twice before starting 2017 and 2018 experiments. We added the information (Line 119-120), and revised the part (Line 357-359).
Your comments: This section needs to explain by how much the manure application would be expected to increase OC and compare it to the 5% lsd. The OC may have increased but by less than the statistical detection limit.
Our answer: The 5% LSD of OC was 0.59, and the value was much lower than expected. As mentioned above, this is probably because of the large amount of CO2 emission from the decomposition of OC in sandy land (Zhenghu et al., 2001). We added the explanation ()Line 354-355.
Your comments: Line 301 “cycle should the crop encounter excess”
Our answer: We revised the part (Line 365-366).
Your comments: Line 302 “values, and had the”
Our answer: We revised the part (Line 367).
Your comments: Line 332 “therefore split applications of N or a slow-release source such as manure might”
Our answer: We revised the part (Line 400-401).
Your comments: Line 334 “furrows were concluded to be” The logic behind this conclusion is not in the Discussion. It would be a useful Discussion point and conclusion
Our answer: We revised the part (Line 403-404), and the logic was described in Line 373-377.